# Bactericidal Efficacy and Mechanisms of Non-Electrolytic Slightly Acidic Hypochlorous Water on *Pseudomonas fragi* and *Pseudomonas fluorescens*

**DOI:** 10.3390/foods12213980

**Published:** 2023-10-31

**Authors:** Qianru Chen, Yanfang Zhou, Xueyan Yun, Namula Zhao, Hongyu Bu, Tungalag Dong

**Affiliations:** 1College of Food Science and Engineering, Inner Mongolia Agricultural University, Hohhot 010018, China; qrchen_nmg@163.com (Q.C.); zhouyf7101@139.com (Y.Z.); yun_imau@163.com (X.Y.); 2Shanghai Wanlay Environmental Technology Co., Ltd., Shanghai 200129, China; 15647113678@139.com; 3Inner Mongolia Institute for Drug Control, West Taoli Street, Hohhot 010020, China; bhongyu163@163.com

**Keywords:** non-electrolytic slightly acidic hypochlorous water, *Pseudomonas fragi*, *Pseudomonas fluorescens*, bactericidal efficacy, mechanisms

## Abstract

Chilled pork is frequently contaminated with *Pseudomonas fragi* and *Pseudomonas fluorescens*. In this study, the bactericidal efficacy and mechanisms of non-electrolytic slightly acidic hypochlorous water (NE-SAHW) against two strains of these two species were evaluated. The results showed that the antibacterial efficacy of NE-SAHW was positively correlated with the concentration level of NE-SAHW and negatively correlated with the initial populations of the strains. The strains of small populations were completely inhibited when provided with each level of NE-SAHW. The killed cells of *P. fragi* were 0.94, 1.39, 4.02, and 5.60 log10 CFU/mL, respectively, and of *P. fluorescens* they were 1.21, 1.52, 4.14, and 5.74 log10 CFU/mL, respectively, when the initial populations of the strains were at high levels (about 7 log10 CFU/mL). Both strains were completely killed within 12 s with the available chlorine concentration (ACC) of 50 mg/L of NE-SAHW. Morphological changes in both cells were observed by using a Scanning Electron Microscope (SEM) and it was discovered that the cell membranes were damaged, which led to the leakage of the intracellular substances, including K+, nucleic acid, and protein. In terms of the Fourier Transform Infrared Spectroscopy (FTIR) results, NE-SAHW destroyed the structures of membrane proteins and cell structure proteins, and influenced the composition of polysaccharides. The bacteria were definitely dead after treatment by NE-SAHW compared to the control according to the results of flow cytometry. These results demonstrated the potential bactericidal property of NE-SAHW when applied to the meat and other food sterilization industries.

## 1. Introduction

Microbial spoilage resulting in postharvest or postslaughter losses can cause the waste of approximately one-third (about 1.3 billion tons annually) of food produced for worldwide human consumption [1]. *Pseudomonas* spp. are mostly aerobic and exist widely in the environment and are the main spoilage bacteria of chilled pork, producing off-flavor compounds, degrading nutritional components and decreasing the shelf life [2]. *Pseudomonas*, a genus of Gram-negative bacteria, are the most common psychrotrophic organisms to cause spoilage [3]. *P. fluorescens* and *P. fragi* are recognized as among the most important spoilage microorganisms due to their short generation time and low-temperature resistance [4].

Essential disinfection processes are necessary and required to control *Pseudomonas* spp. during food production and processing. Sodium hypochlorite (NaClO)-based solutions have been the widely used sanitizers in the food industry in the last decades [5]. However, some disinfection by-products are generated during the disinfection process, such as trihalomethanes and haloacetic acids, which are associated with environmental and human health risks and low chlorine concentrations and cannot achieve the purpose of reducing microbial quantity [5,6].

To inhibit microbial contamination caused by the spoilage of microorganisms and maintain the nutritional benefits and sensory qualities of food, electrolyzed water (EW) has been recognized as holding high disinfection in terms of function among chlorine sanitizers in the food industry [7,8,9]. In general, EW is generated by electrolysis of saline solution (NaCl or KCl) contained within an electrolytic chamber with or without a membrane [10]. Chlorine compounds in EW, including hypochlorous acid (HOCl), Cl_2_, and hypochlorite ions (-OCls), are produced through a series of reactions in the electrolysis system, of which HOCl is mainly responsible for the bactericidal effect [11]. The forms of chlorine compounds in EW are usually influenced by pH, and HOCl becomes the main extant form, with a pH of 5.0–6.5 [12,13].

In this study, the sanitizer used was NE-SAHW (pH 6.50 ± 0.05), which was produced by a non-electrolytic generator, and mixed with NaOCl and HCl in different ratios, which resulted in the main ingredient of HOCl under this pH. The manufacturing processes of NE-SAHW are different from those of slightly acidic electrolyzed water (SAEW), with the same ingredient. As a sanitizer, the disinfection efficacy of HOCl is eighty times more effective than an equivalent level of OCl- in deactivating bacteria [14,15]. Therefore, NE-SAHW contains an equivalent bactericidal potential to SAEW at the same level or under the same available chlorine concentration (ACC). It is worth noting that NE-SAHW is produced through the principle of phase interface reaction, which exhibits a lower cost, without using electrical energy, compared to SEAW, and shows broader application prospects in the food industry. Thus, it is necessary to study the bactericidal efficacy and mechanisms of NE-SAHW. As it is known that SAEW has been approved as a food additive in Korea, Japan, and the US [16], thus, NE-SAHW can be used for the same ingredients as SAEW. Large numbers of studies have applied SAEW for the microbial decontamination of foods, such as postharvest and fresh-cut vegetables, seafood and chilled meat [17,18,19,20,21]. NE-SAHW can also be applied directly on the meat/food surfaces because the ingredients of NE-SAHW are the same as SAEW. It was reported that SAEW exhibited stronger antibacterial efficacy against *Escherichia coli*, *Bacillus subtilis,* and *L. monocytogenes* [22,23,24].

The germicidal mechanisms of SEAW have not been completely elucidated, but a model explaining the germicidal mechanisms of SEAW showed it attacking multiple cellular targets, like the cytoderm, outer membrane, and intracellular components [25,26]. Then, HOCl and reactive oxygen species (ROS) produced from the microbial cell would induce complex intracellular metabolites’ changes, including the inhibition of nucleotide and amino acid biosynthesis, the enhancement of fatty acid metabolism, and the suppression of energy-associated metabolism, like glycolysis and ATP replenishment, or a decrease in the activities of several key enzymes [27,28].

However, there are few studies about the antibacterial effect of NE-SAHW and its germicidal mechanisms, which limits its widespread application. The purpose of this present study was to evaluate the bactericidal efficacy and mechanisms of NE-SAHW on *P. fragi* and *P. fluorescens,* which are the main spoilage organisms of chilled pork, to provide evidence to add to the research on the bactericidal mechanism of NE-SAHW. First, the relationship between the different ACCs and initial bacterial populations of *P. fragi* and *P. fluorescens* were discussed. Subsequently, the cellular membrane integrity of the bacteria was investigated based on Scanning Electron Microscopy (SEM) and the population levels of the bacteria were determined through flow cytometry. Then, the compositions of the cells were detected by Fourier Transform Infrared Spectroscopy (FTIR), and the leakage of intracellular materials, including K+, intracellular nucleic acid, and protein was investigated after treatment by NE-SAHW. This study is very significant in that it promotes the practical application of NE-SAHW and a reduction in the usage of harmful bactericides.

## 2. Materials and Methods

### 2.1. Material, Bacterial Strain, Culture, and Bacterial Suspension Methods

NE-SAHW was prepared by generating equipment (C-5, 723C) from Shanghai Wanlay Environmental Technology Co., Ltd. (Shanghai, China). NE-SAHW was generated by mixing NaOCl and HCl in different ratios by jet flow reaction and regulating pH, and the schematic diagram of a circulating NE-SAHW generation unit is shown in Figure 1. The bacterial strains used were *Pseudomonas fluorescens* (BNCC 336632), *Pseudomonas fragi* (BNCC 134017) which were both obtained from BeNa Culture Collection (Beijing, China). Sodium Caseinate was obtained from Coolaber Technology Co., Ltd. (Beijing, China). The microorganisms were kept frozen at −80 °C and cultivated in nutrient broth (NB) medium at 37 °C for 24 h and then activated for three generations. NB medium: Tryptone 10 g; Beef extract 3 g; Sodium chloride 5 g; Distilled water 1 L; pH 7.0; 121 °C high-pressure sterilization for 15 min before use. NA medium: Tryptone 10 g; Beef extract 3 g; Sodium chloride 5 g; Agar 15 g; Distilled water 1 L; pH 7.0; 121 °C high-pressure sterilization for 15 min before use.

The cultures were centrifuged (2000× *g* for 15 min at 4 °C) using a refrigerated centrifuge and washed twice with sterile 0.85% saline solution. The resulting pellets were re-suspended in phosphate buffer solution to a final concentration of 10^7^, 10^5^, 10^2^, CFU/mL by plate coating counting method.

### 2.2. The Antibacterial Effect of NE-SAHW against P. fragi and P. fluorescens

500 µL of the inocula of 10^7^, 10^5^, 10^2^ CFU/mL bacterial suspension were added to 500 µL solutions with different levels NE-SAHW (10, 20, 30, 40, and 50 mg/L) for 1 min, respectively. Meanwhile, 500 µL of the inocula of 10^7^, 10^5^, 10^2^ CFU/mL bacterial suspension were added to 500 µL of NE-SAHW at 50 mg/L for 0.2, 1, 5, and 10 min, respectively. Reactions were stopped by a terminating agent (0.5% Na_2_S_2_O_3_, 0.85% NaCl). After gradient dilution, the amount of cells surviving were counted by direct plating of 0.1 mL of each dilution on NA medium. The petri dishes were incubated at 37 °C for 48 h. The bacteria counts were expressed in log10 CFU/mL.

### 2.3. Observation of Cell Membrane by SEM

1 mL of the inocula of 10^6^ CFU/mL of the two bacteria were added to 1 mL 50 mg/L NE-SAHW solutions for 1 min, respectively. The suspensions were centrifuged at 7104× *g* at 4 °C for 10 min, and sediments were collected and washed three times using sterile PBS. The precipitates were fixed with 2.5% glutaraldehyde for 5 h, and then dehydrated in a graded ethanol series (30, 50, 70, 80, 90, 95, and 100%) for 10 min. Critical point drying with liquid CO_2_ was conducted on the samples and observed under SEM (TM 4000; HITACHI, Tokyo, Japan).

### 2.4. Observation of Cell Membrane by FTIR

The molecular composition changes of bacteria were analyzed by FTIR according to a previous study with some modifications [29]. Amounts of 1 mL of the inocula of 10^6^ CFU/mL of the two bacteria were added to 1 mL 50 mg/L NE-SAHW solutions for 1 min, respectively, and Dimethyl sulfoxide (DMSO) was added in the control sample. All the samples were incubated at 37 °C for 6 h and then centrifuged at 7104× *g* at 4 °C for 10 min. The sediments were collected and washed three times with sterile PBS. After freeze-drying, the samples were mixed with spectroscopically pure potassium bromide. Then, the transparent sheets were prepared and detected under FTIR (IRAffinity-1; Shimadzu, Tokyo, Japan) by attenuated total reflection (ATR) through the following parameters: wavenumbers ranging from 4000–500 cm^−1^, resolutions at the ratio of 4 cm^−1,^ and scanning number was 64 times.

### 2.5. Leakage of Bacterial Cellular Materials (Protein, Nucleic Acid, and K+)

1 mL of the inocula of 10^6^ CFU/mL of the two bacterial suspensions were added to 1 mL solutions with different levels of NE-SAHW (10, 30, and 50 mg/L) for 1 min, respectively. The samples were cultured at 30 °C for 3 h and then centrifuged at 7104× *g* at 4 °C for 10 min. The supernatants were used to take further measurements. The concentrations of protein and nucleic acid were determined by a UV-VIS spectrophotometer (UV 2450, Shimadzu Corp, Japan) at wavelengths of 280 and 260 nm, respectively. K+ content was measured by K+ concentration assay kit (C001-2-1, Nanjing Jiancheng Bioengineering Institute, Nanjing, China). The same treatments with sterile saline were used as control.

### 2.6. Determination of Bacterial Death by Flow Cytometry

Determination of bacterial death of two strains was assessed using propidium iodide (PI)-stained flow cytometry. PI is a fluorescent dye that can only penetrate through dead/damaged cells and binds with nucleic acid, which exhibits red fluorescence. Amounts of 1.5 mL of inocula of 10^6^ CFU/mL of the two bacterial suspensions were prepared, and centrifuged at 7104× *g* at 4 °C for 10 min. The supernatant was discarded, and the pellet was resuspended with aseptic PBS buffer three times and treated with 50 mg/L NE-SAHW. The control group consisted of the bacterial solution without NE-SAHW. The cell samples fixed with 20 µL PI were analyzed using a flow cytometer (CytoFlex, You Hua, Beijing, China), and the emission of PI dye was 488 nm. The data were analyzed using CellQuest Pro software Version 5.1 (BD Biosciences, San Jose, CA, USA).

## 3. Results and Discussion

### 3.1. Bactericidal Efficacy of NE-SAHW on the Strains of P. fragi and P. fluorescens

The bactericidal efficacy of different levels of NE-SAHW and different treatment times on strains of *P. fragi* and *P. fluorescens* were evaluated in this present study, respectively, and the results are shown in Table 1 and Table 2 and Figure 2 and Figure 3. The initial low, medium, and high population levels of the strains were about 2, 5, 7 log10 CFU/mL, respectively. As shown in Table 1 and Table 2, the strains with low population levels were completely inhibited when provided with each level of NE-SAHW, which can also be verified in Figure 2 and Figure 3, as these results demonstrated the strong antibacterial efficacy of NE-SAHW against the two strains. As the results show in Table 1, when the initial populations of the strains were about 5 log10 CFU/mL, killed cells of *P. fragi* were 1.99 and 3.93 log10 CFU/mL, which resulted from NE-SAHW concentrations of 10 mg/L and 20 mg/L, respectively, and the fatality rate was 38.3% and 75.6% which showed the same trend as *P. fluorescens*. When the initial populations of the strains were about 7 log10 CFU/mL, the bactericidal efficacy was enhanced with the increased level of NE-SAHW. The killed cells of *P. fragi* were 0.94, 1.39, 4.02, and 5.60 log10 CFU/mL, respectively, and of *P. fluorescens* they were 1.21, 1.52, 4.14, and 5.74 log10 CFU/mL, respectively. According to the results shown in Table 2, NE-SAHW at 50 mg/L had a significant disinfection effect, by which the strains of *P. fragi* and *P. fluorescens* were completely killed within 12 s. The bactericidal efficacy was enhanced with the treatment time when the initial level of the strains was relatively high (about 7 log10 CFU/mL) and *P. fragi* and *P. fluorescens* killed cells were 5.60 and 5.74 log10 CFU/mL (78.7% and 82.5%), respectively.

The sharp bactericidal efficacy of SAHW has been demonstrated by many researchers, and SAHW has exhibited that it is a promising forerunner in the food industry [7,8,30]. The present study showed its strong bactericidal efficacy, which agrees with previous reports [17,22,23]. As the results show, even the same bacteria genus, when exposed to the same concentration of NE-SAHW, exhibited obvious differences in sensitivity to NE-SAHW. Fenner et al. (2006) [31] reported that *Proteus mirabilis* and *S. aureus* were more sensitive than *Mycobacterium avium subsp. avium*, *Enterococcus faecium*, and *Pseudomona saeruginosa* to EW. Therefore, more attention should be paid to the different sensitivities of microorganisms to NE-SAHW when it is used in food preservation.

### 3.2. Effect of NE-SAHW on Cell Membrane of P. fragi and P. fluorescens by SEM

Based on the strong bactericidal efficacy of NE-SAHW, the damage to the cell membrane of the two strains was determined to explain the mechanism of NE-SAHW in one aspect. The cell membrane is an important component of a cell; it protects the cell from external damage and contains the function of controlling material exchange. SEM images confirmed that *P. fragi* and *P. fluorescens* subjected to NE-SAHW treatment were morphologically changed (Figure 4). The control samples of both cell bodies were uniformly distributed and exhibited complete morphology, smooth surfaces, and regular texture (Figure 4A,C). After treatment with NE-SHAW with an ACC of 50 mg/L, bacterial cells were partially damaged and emerged shrunken and partially fractured (Figure 4B,D). The cell membrane may be damaged by the strong oxidizing effect of NE-SAHW, causing the release of intracellular material and making the osmotic pressure out of balance. Similarly, Hao et al. (2017) [32] reported that remarkable changes in the cell membrane of *Escherichia coli*, *Staphylococcus aureus*, and *Bacillus subtilis* were observed after treatment with NE-SAHW by SEM.

### 3.3. Effect of NE-SAHW on Intracellular Material Leakages of P. fragi and P. fluorescens

Intracellular material leakages of substances such as protein, nucleic acids, and K+ reflect the alterations in the cell membrane’s permeability, and the damage to the cell membrane should lead to bacterial death [33]. As shown in Figure 5A, the protein and nucleic acid levels of *P. fragi* increased significantly after NE-SAHW treatment in a concentration-dependent manner. Following NE-SAHW with an ACC of 50 mg/L, OD_280_ was 0.42, which was 133.3% higher than the untreated control cells (OD_280_ of the control: 0.18, *p* < 0.05). Meanwhile, following NE-SAHW with an ACC of 30 mg/L, OD_260_ was 0.47, which was 88% higher than the untreated control cells (OD_280_ of the control: 0.25, *p* < 0.05). The interesting thing is, with NE-SAHW with an ACC of 15 mg/L, K+ levels were increased to 0.23 mmol/L, which showed the same strong leakage as the result of the 30 mg/L and 50 mg/L treatment, and this phenomenon’s occurrence may be attributed to the damage caused by NE-SAHW to the cell membrane and the original small molecular weight leakage. As the NE-SAHW level increased, the content of the cell membrane damage became more serious. Similar change trends were also observed for the leakage of the intracellular material of *P. fluorescens* (Figure 5B). These results are in agreement with Li et al. (2021) [34], who found that the leakage of intracellular protein and the level of ACC in SAEW after treatment presented a certain degree of positive correlation.

### 3.4. Effect of NE-SAHW on Molecular Composition Changes in Bacteria

The mechanisms of bacterial death induction can be illuminated by FTIR spectroscopy after exposure to different antimicrobial compounds or changes of environmental conditions [35]. Molecular composition changes in bacteria can be reflected, including cytoplasmic proteins, cell membrane phospholipids, cell wall polysaccharides, and nucleic acids [36]. The wavenumbers between 3000 and 2800 cm^−1^ reflect lipid regions assigned to the alkyl group of lipids [37]. The bands at 1650, 1546, 1458, 1395 cm^−1^ correspond to protein features, along with the band at 1650 cm^−1,^ assigned to amide I of the b-pleated sheet, 1545 cm^−1,^ assigned to amide II, the N-H bonding of amide of proteins, and 1458 and 1395 cm^−1,^ assigned to cell structure proteins, respectively [38,39]. The bands at 1235, 1080, 1200, and 900 cm^−1^ correspond to P=O asymmetric stretching (PO_2_- phosphodiesters), nucleic acids, and polysaccharides, respectively [36,37].

As shown in Figure 6 and Figure 7, after treatment with NE-SAHW, the spectra features of both *P. fragi* and *P. fluorescens* exhibited obvious changes compared with the control. For *P. fragi*, the spectra features showed no difference in 3000~2800 cm^−1^ (lipids), 1235 cm^−1^ (phosphodiesters), and 1080 cm^−1^ (nucleic acids), representing regions which indicated that NE-SAHW cannot damage the cell membrane phospholipids and nucleic acids. In contrast, in the protein area, the spectra features coincided or crossed at 1637 cm^−1^ (amide I of the b-pleated sheet), the shoulder peak was lost at 1545 cm^−1^ (amide II of the N-H bonding) and 1458 cm^−1^ (cell structure proteins), compared with the control, and on the whole, the changes indicated that NE-SAHW can change the structures of membrane proteins and cell structure proteins. Compared with *P. fragi*, similar phenomena were also observed in *P. fluorescens,* although some different performances existed. The spectra features of the NE-SAHW treatment appeared to bulge and concave at the band of 1200–800 cm^−1,^ which indicated that NE-SAHW influenced the composition of the polysaccharides. The potential mechanisms for bacterial survival could be to regulate down most functions to save energy to produce polysaccharides and oligosaccharides for protection against environmental damage [40].

### 3.5. Determination of Bacterial Death by Flow Cytometry

As a fluorescent dye, propidium iodide (PI) can only penetrate through dead/damaged cell walls and bind with nucleic acid, which should exhibit red fluorescence, whereas it cannot enter inside viable cells, as the cell membrane remains intact in viable cells, which would show no fluorescence [41]. Thus, the amount of PI interaction with nucleic acid signifies cell wall damage or perforation [42]. In this study, the bacterial death of *P. fragi* and *P. fluorescens* cells was analyzed by flow cytometry, as shown in Figure 8. The results showed that 99.23% of the *P. fragi* had intact cell membranes and 0.77% of the cells were colored by PI (Figure 8A), indicating that 0.77% of the cells were dead, probably because of the mechanical damage. However, after treatment with the dose of 50 mg/L NE-SAHW, a large proportion of the cells were transformed from having a low area to a high area of PI coloration, and the proportion of the *P. fragi* low-area coloration decreased to 42.1% (Figure 8B), revealing that 57.9% of the cells were damaged by NE-SAHW. Similarly, the proportion of *P. fluorescens* dead cells was 57.9% when treated with the dose of 50 mg/L NE-SAHW, compared with the control group’s proportion of 0.52% (Figure 8C,D), indicating the strong antibacterial effect of NE-SAHW as well.

## 4. Conclusions

In this present work, the bactericidal efficacy and mechanisms of NE-SAHW against *P. fragi* and *P. fluorescens*, the main spoilage organisms of chilled pork, have been conducted and evaluated. These results indicated that NE-SAHW exhibited a bactericidal effect against the two strains. The ACC of NE-SAHW, treatment time, and initial bacterial populations played important parts in the bactericidal efficacy. The ACC of NE-SAHW after treatment presented a certain degree of positive correlation. The results of the bactericidal mechanisms demonstrated that NE-SAHW could destroy the cell membrane of *P. fragi* and *P. fluorescens*, thus leading to the leakage of cellular substances including K+, protein, and nucleic acid. Furthermore, NE-SAHW could disrupt the structures of membrane proteins and cell structure proteins, and influenced the composition of polysaccharides. The bacteria were definitely dead after treatment by NE-SAHW compared to the control according to the results of flow cytometry. These results provided evidence for the bactericidal property of NE-SAHW against *P. fragi* and *P. fluorescens* and of some bactericidal mechanisms.

## Figures and Tables

**Figure 1 foods-12-03980-f001:**
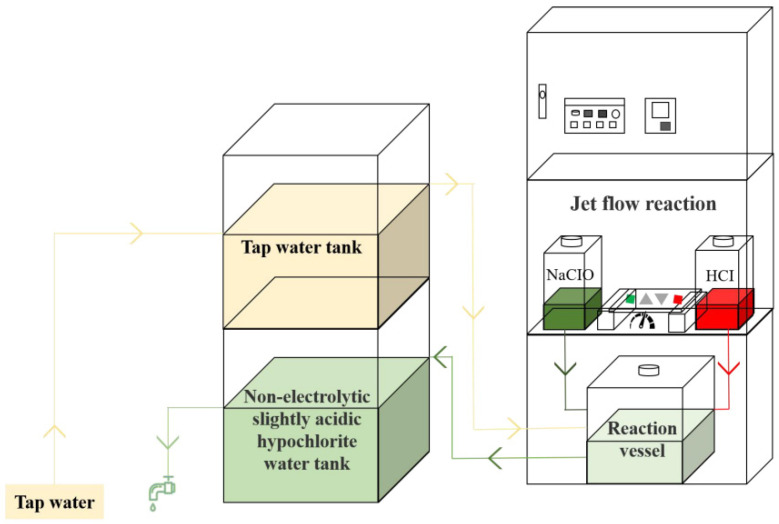
Schematic diagram of a circulating NE-SAHW generation unit.

**Figure 2 foods-12-03980-f002:**
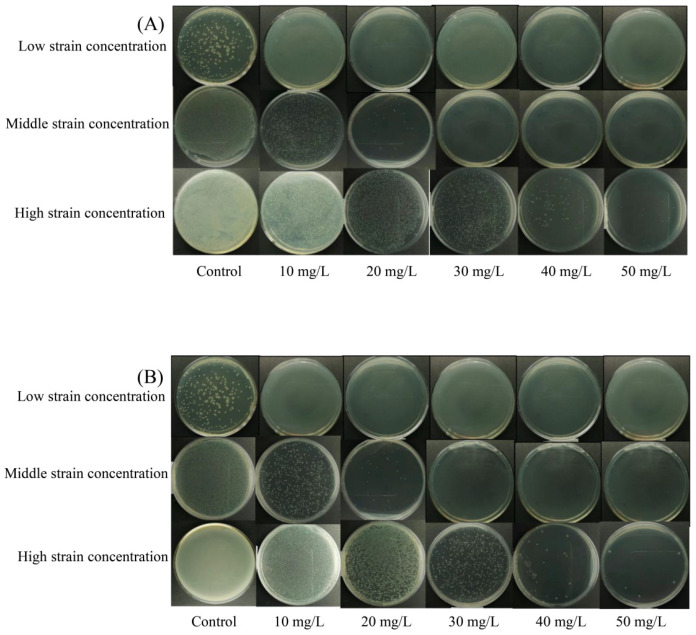
Experiment diagram of bactericidal efficacy of NE-SAHW with different ACCs (10, 20, 30, 40, 50 mg/L) on the two strains: (**A**) *P. fragi*, (**B**) *P. fluorescens*.

**Figure 3 foods-12-03980-f003:**
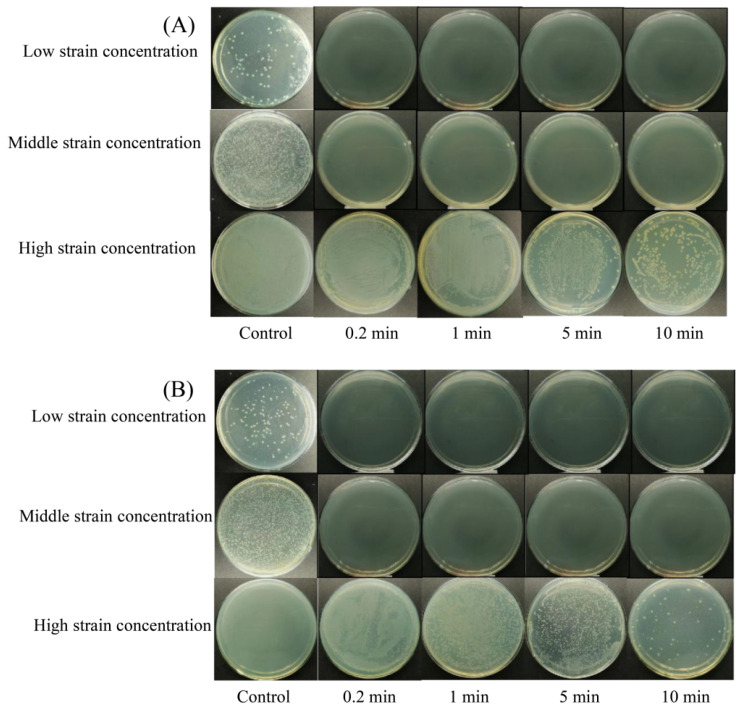
Experiment diagram of bactericidal efficacy of NE-SAHW with different treatment times (0.2, 1, 5, 10 min) on the two strains: (**A**) *P. fragi*, (**B**) *P. fluorescens*.

**Figure 4 foods-12-03980-f004:**
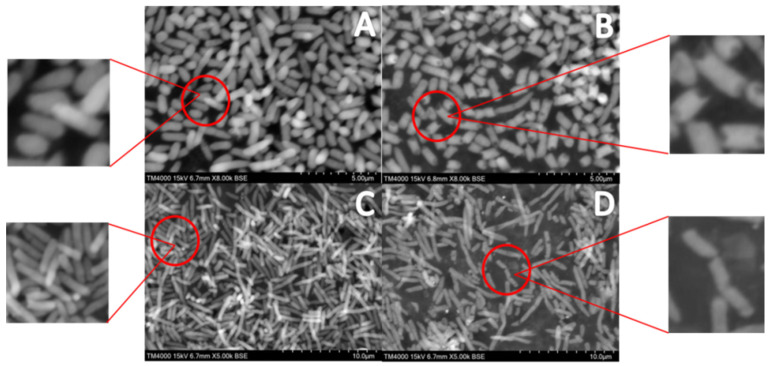
SEM of *P. fragi* and *P. fluorescens*: (**A**) *P. fragi* without treatment by NE-SAHW, (**B**) *P. fragi* treated by NE-SAHW (ACC of 50 mg/L), (**C**) *P. fluorescens* without treatment by NE-SAHW, (**D**) *P. fluorescens* treated by NE-SAHW (ACC of 50 mg/L).

**Figure 5 foods-12-03980-f005:**
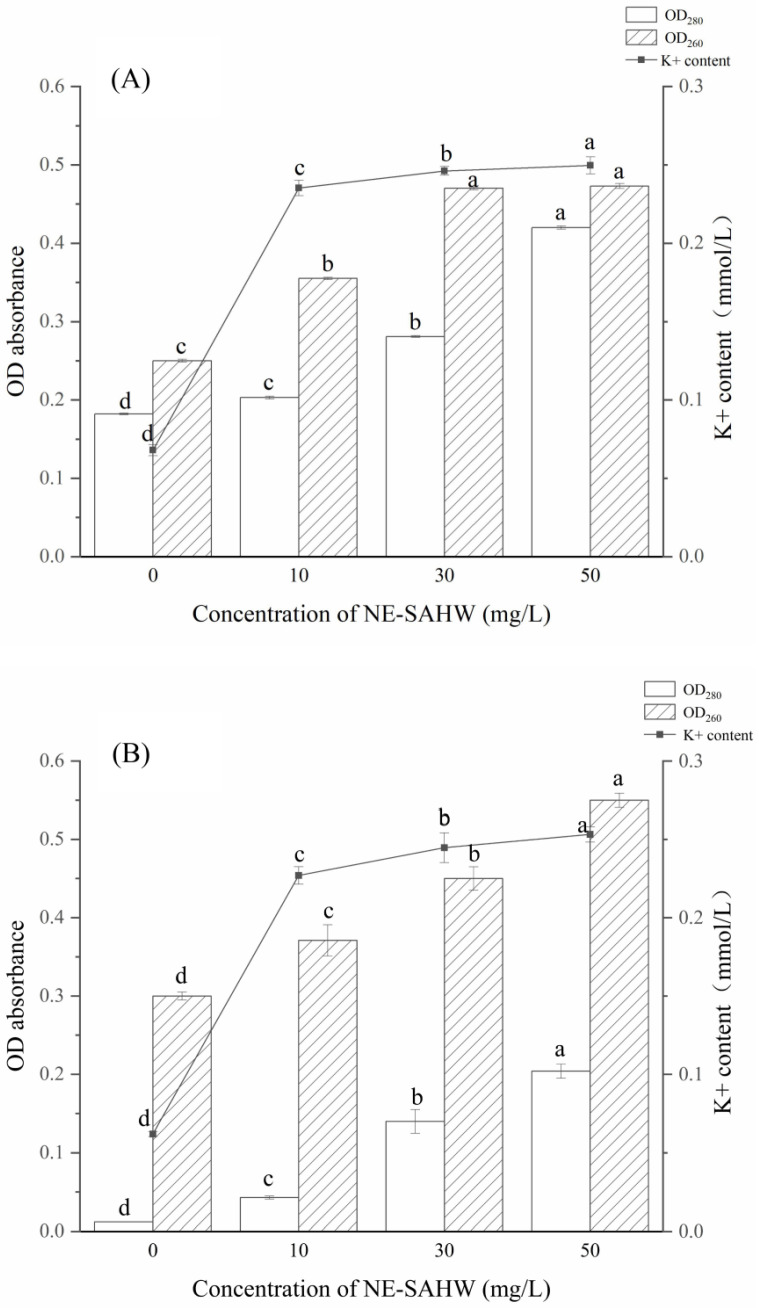
Effect of NE-SAHW on the leakage of cellular materials of the two strains including K+, cellular protein, and nucleic acid. (**A**) *P. fragi*, (**B**) *P. fluorescens*. The data are presented as mean standard deviation (SD). Different letters on the bar mean the difference between samples (*p* < 0.05).

**Figure 6 foods-12-03980-f006:**
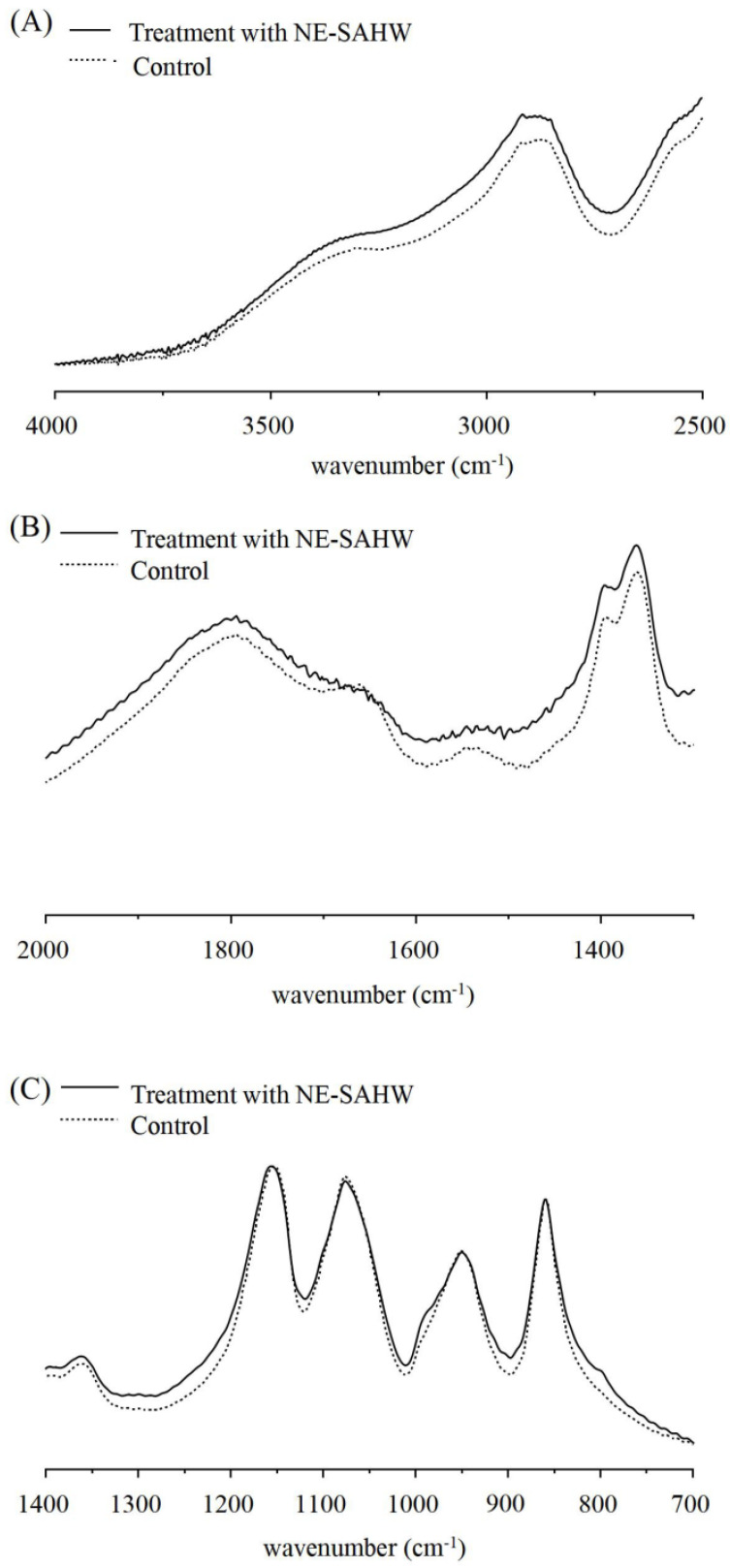
FTIR spectra of *P. fragi* with and without the ACC of 50 mg/L of NE-SAHW from the wavenumber at (**A**) 4000 to 2500 cm^−1^, (**B**) 2000 to 1400 cm^−1^, (**C**) 1400 to 700 cm^−1^, respectively.

**Figure 7 foods-12-03980-f007:**
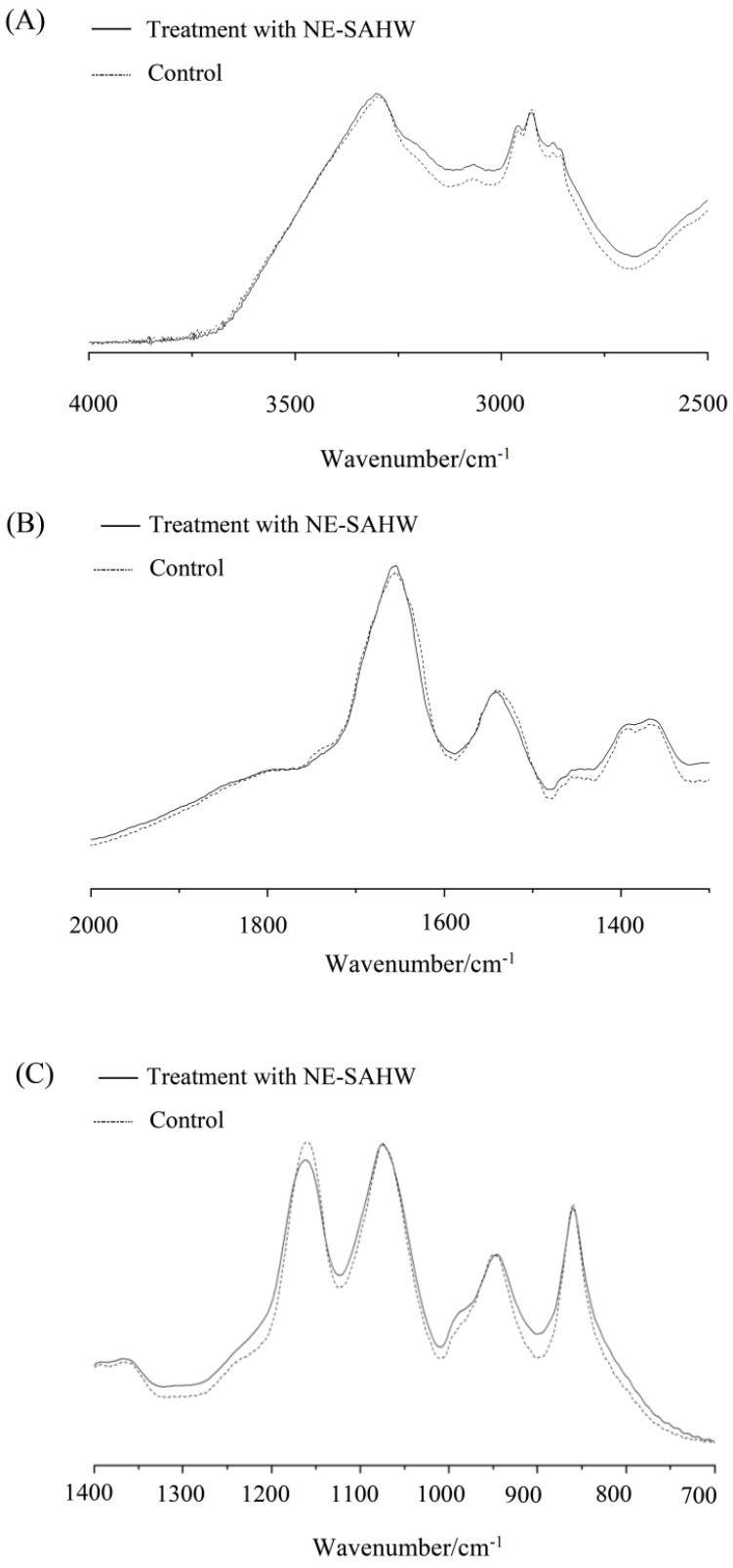
FTIR spectra of *P. fluorescens* with and without NE-SAHW with an ACC of 50 mg/L from the wavenumber at (**A**) 4000 to 2500 cm^−1^, (**B**) 2000 to 1400 cm^−1^, (**C**) 1400 to 700 cm^−1^.

**Figure 8 foods-12-03980-f008:**
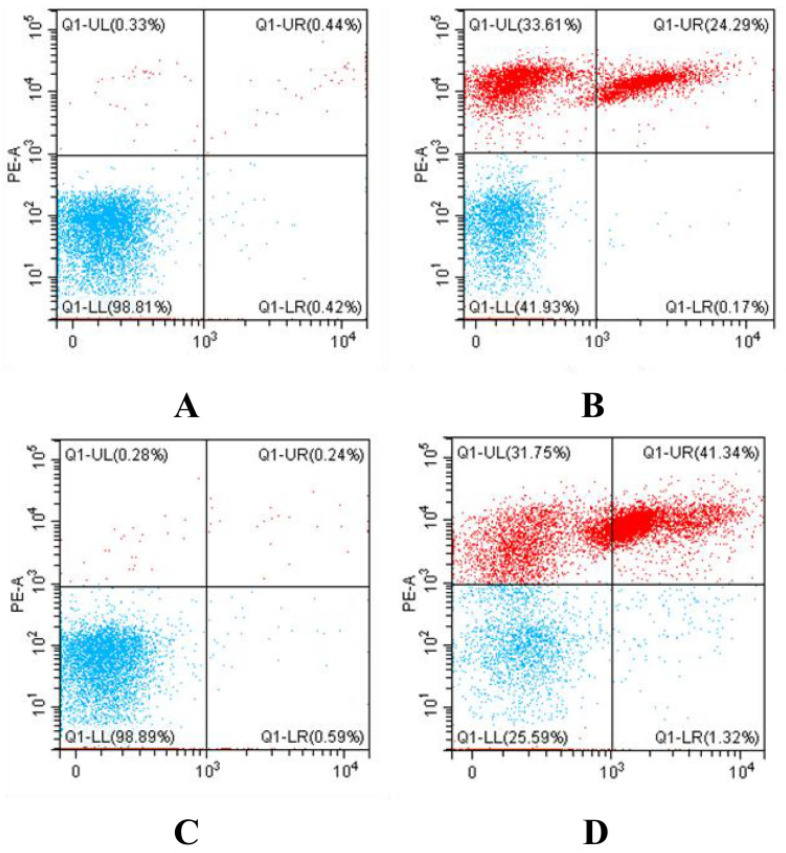
Flow cytometry of *P. fragi* and *P. fluorescens*: (**A**) *P. fragi* without treatment by NE-SAHW, (**B**) *P. fragi* treated by NE-SAHW (ACC of 50 mg/L), (**C**) *P. fluorescens* without treatment by NE-SAHW, (**D**) *P. fluorescens* treated by NE-SAHW (ACC of 50 mg/L).

**Table 1 foods-12-03980-t001:** Bactericidal efficacy of NE-SAHW at different levels on strains of *P. fragi* and *P. fluorescens*.

Initial Populations of the Strains		Concentration of NE-SAHW (mg/L)	Log Reduction (log10 CFU/mL)	
			*P. fragi*	*P. fluorescens*
Low population levels	*P. fragi* 2.27 log10 CFU/mL *P. fluorescens* 2.08 log10 CFU/mL	10	>1.97	>1.81
		20		
		30		
		40		
		50		
Medium population levels	*P. fragi* 5.20 log10 CFU/mL *P. fluorescens* 5.06 log10 CFU/mL	10	1.99	1.96
		20	3.93	3.98
		30	>4.9	>4.76
		40		
		50		
High population levels	*P. fragi* 7.23 log10 CFU/mL *P. fluorescens* 7.05 log10 CFU/mL	10	1.83	1.92
		20	3.5	3.17
		30	4.38	4.35
		40	5.56	5.78
		50	6.16	6.23

**Table 2 foods-12-03980-t002:** Bactericidal efficacy of NE-SAHW with different treatment times on strains of *P. fragi* and *P. fluorescens*.

Initial Level of the Strains		Treatment Time (min)	Log Reduction (log10 CFU/mL)	
			*P. fragi*	*P. fluorescens*
Low population levels	*P. fragi* 2.36 log10 CFU/mL *P. fluorescens* 2.38 log10 CFU/mL	0.2	>2.16	>2.08
		1		
		5		
		10		
Medium population levels	*P. fragi* 5.23 log10 CFU/mL *P. Fluorescens* 5.06 log10 CFU/mL	0.2	>4.93	>3.76
		1		
		5		
		10		
High population levels	*P. fragi* 7.12 log10 CFU/mL *P. fluorescens* 6.96 log10 CFU/mL	0.2	0.94	1.21
		1	1.39	1.52
		5	4.02	4.14
		10	5.60	5.74

## Data Availability

Data is contained within the article.

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
