# Peer review of "Bactericidal Efficacy and Mechanisms of Non-Electrolytic Slightly Acidic Hypochlorous Water on Pseudomonas fragi and Pseudomonas fluorescens"

_foods, 2023, doi:10.3390/foods12213980_

Round 1

Reviewer 1 Report

Comments and Suggestions for Authors

The evaluated study described eficacy of bactericidal effect of non-electrolytic slightly acidic hypochlorous water (NE-SAHW) on P. fragi and P. fluorescens strains. The authors treated a different inocula with various level of the mentioned strains with different concentrations of NE-SAHW. The researchers checked the effect on bacterial growth, as well as cell membrane of P. fragi and P. fluorescens permeability, also molecular composition changes of bacteria and bacterial death by flow cytometry was investigated. the manuscript describes the molecular mechanism of changes at the cellular level under the influence of NE-SAHW. This knowledge can be very useful from the point of view of food technologists dealing with aspects of microbiological preservation of meat.

Underneath are some remarks regarding doubts related to some phrases.

The title sugests that mentioned strains of P. fragi and P. fluorescens originated from pork meat. Are the strains used in the study (BNCC 336632 and BNCC 134017) have been isolated from pork meat? - please explain. At the begining has been mentioned that such a strains are occassional contaminants of pork meat and in the conclusion the same strains are determined as  "main pathothens" - please clarify.

Some abreviations should be explained in the brackets at the begining of the study, such as:

-line 20 - ACC;

-L. 21 -  SEM;

-L. 23- FTIR

The last sentence in the introduction: "This study is very significant to promote practical application of NE-SAHW and to reduce the usage of the harmful fungicides." - is this correct notice? This study has'nt described fungicidal effect of NE-SAHW.

L. 122 - Please define LB medium of address producer data

L. 155 - define PI-stained flow cytometry

L. 137, 147, 157 - please change r/min. into "RCF" or "x g"

Author Response

The title sugests that mentioned strains of P. fragi and P. fluorescens originated from pork meat. Are the strains used in the study (BNCC 336632 and BNCC 134017) have been isolated from pork meat? - please explain. At the begining has been mentioned that such a strains are occassional contaminants of pork meat and in the conclusion the same strains are determined as "main pathothens" - please clarify.

The two strains were purchased from BeNa Culture Collection (Beijing, China) and we did not isolated them from pork meat. The two strains are the main pathogen of pork which the references afforded could be proved this. Because the two strains are the main pathogens of the chilled pork, the inhibition against the two strains by using NE-SAHW could reflect the influence of NE-SAHW on chilled pork preservation. The word “occasionally” is wrong and it should be modified as “frequently”, I am so sorry that this is a language error. 

Some abreviations should be explained in the brackets at the begining of the study, such as:

-line 20 - ACC;

It has been explained in the manuscript.

-L. 21 -  SEM;

It has been explained in the manuscript.

-L. 23- FTIR

It has been explained in the manuscript.

The last sentence in the introduction: "This study is very significant to promote practical application of NE-SAHW and to reduce the usage of the harmful fungicides." - is this correct notice? This study has'nt described fungicidal effect of NE-SAHW.

It has been corrected in the manuscript.

  1. 122 - Please define LB medium of address producer data

It should be NA medium which are also explained in material and method (2.1. Material, bacterial strain, culture and bacterial suspension methods), I am so sorry about this language mistake.

  1. 155 - define PI-stained flow cytometry

It has been defined.

  1. 137, 147, 157 - please change r/min. into "RCF" or "x g"

It has been corrected.

Reviewer 2 Report

Comments and Suggestions for Authors

The authors describe an alternative sanitation method for pork. They demonstrate the antimicrobial efficacy of non-electrolytic slightly acidic hypochlorous water. The authors also discuss possible underlying mechanisms. Albeit the headline and the abstract promise deeper insides into the mechanisms, the authors haven´t provide any experiments in these terms. It is a bit misleading. However, the novelty of the manuscript is given by the sanitation method combined with the target organisms. The manuscript has no serious flaws but needs some further editing. Since the authors need to edit their graphs, I recommend major revisions. 

Comments on the Quality of English Language

Moderate editing of English language required

Author Response

Review: Bactericidal efficacy and mechanisms of non-electrolytic slightly acidic hypochlorous water on Pseudomonas fragi and Pseudomonas fluorescence of chilled pork strains.

 The authors describe an alternative sanitation method for pork. They demonstrate the antimicrobial efficacy of non-electrolytic slightly acidic hypochlorous water. The authors also discuss possible underlying mechanisms. Albeit the headline and the abstract promise deeper insides into the mechanisms, the authors haven´t provide any experiments in these terms. It is a bit misleading. However, the novelty of the manuscript is given by the sanitation method combined with the target organisms. The manuscript has no serious flaws but needs some further editing. Since the authors need to edit their graphs, I recommend accepting the paper after major revisions.

Remarks for the authors:

Line 18: You are using the logarithmic representation for your bacterial count. Please highlight the type of logarithm in a subscript (log10). Correct it throughout your manuscript.

I have correct them all.

Line 63: I don´t understand your sentence. Did you compare your method with another sanitizer, which has been reported by Olimat and Holley (2012) as well as Xuan et al (2016)? Please explain in detail.

Both the two references proposed the opinion and proved that the disinfection efficacy of HOCl is eighty times more effective as a sanitizer than an equivalent level of the OCl- in the inactivation of bacteria.

Line 66: Did you introduce the abbreviations ACC and SAEW? If not, please do so. Albeit they may very familiar abbreviations in your field of research.

I have introduced them in the manuscript.

Line 171: Do the percentage numbers giving a comparable statement than the log10-numbers? Unify!

It has been unified.

Line 200: Did you discuss differences in NE-SAHW action in terms of bacterial envelope proteins or the difference in action on Gram-positive or negative bacteria?

We discussed differences in NE-SAHW action against the two strains which are the main pathogens of chilled pork and compared with the inhibition effect with other research. Thank you so much for your advise and we will discuss NE-SAHW action in terms of bacterial envelope proteins or the difference in action on Gram-positive or negative bacteria in the future research.

Line 205: Did you compare the bactericidal effect of your NE-SAHW with contemporary used sanitizers? It would be interesting to have a comparison.

Thank you so much for your advise, we will do that work in the future. In this manuscript, we focus on the inhibition effect on the main pathogen of chilled pork and its mechanisms.

Line 229: Does the concentration of proteins and nucleic acid increases inside or outside the cell? General remarks: Please improve the caption of your figures. Add information. For instance, add the initial bacterial load in Table 1. It gives the reader an overview and helps to classify the strength of your NE-SAHW as a sanitizer. Did some bacteria possibly enter VBNC? Or why do you need several methods to prove the success of your sanitation step? Discuss and explain! I recommend publishing the manuscript in your journal „foods” after major revisions.

This is the increasing of OD280 and OD260 which could reflect the concentration of protein and nucleic acid but not the position. The the initial bacterial load were showed in table 1. This manuscript has did the major revisions according your advise.

Reviewer 3 Report

Comments and Suggestions for Authors

The article by Chen et al., concerns the bactericidal efficacy of non-electrolytic slightly acidic hypochlorous water (NE-SAHW) against two Pseudomonas strains.

Food scientists and the food industry always welcome new data concerning effective disinfection processes. This research focuses on the use of NE-SAHW against reference strains of Pseudomonas and tries to further elucidate the mode of action of NE-SAHW. There may be a considerable amount of work in this study but this is not depicted in the manuscript. My concerns are critical since I’m not sure that the methodology used and the quality of data presented, are enough to support such conclusions.

At first there is a major consideration about the title. Authors indicate that the two Pseudomonas strains were isolated or somehow related to chilled pork. But, in the manuscript and particularly in M&M section there is not anything relevant. In fact, the word “pork” is mentioned in the introduction and in conclusions.

Second are the abbreviations used in the text. They should be fully defined as soon as they appear in the text. Some are defined later in the text and some (e.g ACC, SAEW) are not defined at all although they are used frequently.

Line 58. check for typos

Line 83: check for typos

Line: 87: was instead of were

Line 87-97: This text doesn’t belong in introduction.

Third is the Materials & Methods:

More details should be provided for us to comprehend the methodology used during the experimentation. For example, “NE-SAHW were generated by mixing NaOCl and  HCl in different ratio by jet flow reaction and regulating pH, and the schematic diagram of a circulating NE-SAHW generation unit was shown as Fig. 1.” is the only information the authors give regarding the NE-SAHW production/generation. For me, and those who might read the article, that info (text and figure) is not sufficient.  Is this a valid method of NE-SAHW production? Has it been used and evaluated before? Are there any reference(s) to support the selection of this method – equipment? Also, there is no relevant info in the internet for the manufacturing company and therefore I’m not willing to approve unknown procedures or equipment built for other uses than for food processing.   

L104-105: “Pseudomonas fluorescens (BNCC 336632), Pseudomonas fragi (BNCC 134017) which were both obtained from BeNa Culture Collection (Beijing, China)”. Again, how these strains are related to frozen pork as indicated in the title and presented in the introduction.

L107-108: “..and then activated three generations”. What do the authors mean? Please describe adequately every procedure you have employed.

L111-112: “The resulting pellets were re-suspended in phosphate buffer solution to a final concentration of 10E7, 10E5, 10E2 log CFU/mL”. How this was accomplished?

L120: “Reactions were stopped by a terminating agent (0.5% Na2S2O3, 0.85% NaCl)”. Again, where this is coming from?

L150-151: “OD280 and OD260 value, respectively, with respectively UV spectrophotometer.” This sentence should be deleted.

L155: “Determination of bacterial death of two bacteria was assessed using PI-stained flow”. You mean two strains and not just two bacteria. Also, PI stands for propidium iodine. Why don’t you explain that?

Fourth are the Results and Discussion (not discussions L 163).

L165-167: “Bactericidal Efficacy of with different level of SAEW and different treatment time on Strains of P. fragi and P. fluorescens were evaluated in this present study respectively and the results were shown in Table 1, 2 and Fig. 2, 3.”

and  L170”.. which could also be verified in Fig. 2, 3..”.

Only numerical figures (data) can support an assumption. Pictures of Petri dishes do not verify anything. The data in Table 1 and 2 are two few to support the findings. These numerical figures also haven’t been compared or statistically verified. How many times did the team repeat the experiments? Where are those data?  Which is the limit of detection for the bacterial enumeration and how did they perform this, with decimal dilutions, plating on agar surface and incubation or the McFarland approach?

L189 & L494: In figure caption “..Experimental diagram of.. ”. In what way a series of petri dish photos are comprising an experimental diagram? There are proper procedures to analyze photos and extract proper information and data to further explore.

L225. Section 3.3. All of your findings should be supported by numerical data from adequate repetitions of experiments.  Those data should be presented mainly in tables and analyzed statistically to support any differences (or similarities) among the various treatments.

L247. Section 3.4

No data to support any assumption. Just unprocessed FTIR spectra. To the common eye, there are no differences between the control and the treatment with NE-SAHW. How many times have you repeated the analysis?

L 282-296. Section 3.5.

In the text, some percentages are discussed (% of intact cells vs. dead). Where are these data in figures A-D?

Comments on the Quality of English Language

minor to moderate editing of English language required

Author Response

The article by Chen et al., concerns the bactericidal efficacy of non-electrolytic slightly acidic hypochlorous water (NE-SAHW) against two Pseudomonas strains.

Food scientists and the food industry always welcome new data concerning effective disinfection processes. This research focuses on the use of NE-SAHW against reference strains of Pseudomonas and tries to further elucidate the mode of action of NE-SAHW. There may be a considerable amount of work in this study but this is not depicted in the manuscript. My concerns are critical since I’m not sure that the methodology used and the quality of data presented, are enough to support such conclusions

At first there is a major consideration about the title. Authors indicate that the two Pseudomonas strains were isolated or somehow related to chilled pork. But, in the manuscript and particularly in M&M section there is not anything relevant. In fact, the word “pork” is mentioned in the introduction and in conclusions.

The two strains are the main pathogen of pork which the references afforded could be proved this. Because the two strains are the main pathogens of the chilled pork, the inhibition against the two strains by using NE-SAHW could reflect the influence of NE-SAHW on chilled pork preservation.

Second are the abbreviations used in the text. They should be fully defined as soon as they appear in the text. Some are defined later in the text and some (e.g ACC, SAEW) are not defined at all although they are used frequently.

They have been defined as soon as they appear in the manuscript.

Line 58. check for typos

It has been revised.

Line 83: check for typos

It has been revised.

Line: 87: was instead of were

It has been revised.

Line 87-97: This text doesn’t belong in introduction.

It has been revised.

Third is the Materials & Methods:

More details should be provided for us to comprehend the methodology used during the experimentation. For example, “NE-SAHW were generated by mixing NaOCl and  HCl in different ratio by jet flow reaction and regulating pH, and the schematic diagram of a circulating NE-SAHW generation unit was shown as Fig. 1.” is the only information the authors give regarding the NE-SAHW production/generation. For me, and those who might read the article, that info (text and figure) is not sufficient.  Is this a valid method of NE-SAHW production? Has it been used and evaluated before? Are there any reference(s) to support the selection of this method – equipment? Also, there is no relevant info in the internet for the manufacturing company and therefore I’m not willing to approve unknown procedures or equipment built for other uses than for food processing.   

The manufacturing equipment was bought from Shanghai Wanlay Environmental Technology Co., Ltd and it is the equipment dedicated to produce non-electrolytic slightly acidic hypochlorous water and this machine was also sold to others to produce NE-SAHW. This producing method uses the principle of phase interface reaction which is a valid and proven approach. There are detail information of the company on the internet, it was founded in 2016 and engaged in environmental protection technology, disinfectant sales, et al. Based on the above evidence, NE-SAHW producing method is correct and trustworthy.  

L104-105: “Pseudomonas fluorescens (BNCC 336632), Pseudomonas fragi (BNCC 134017) which were both obtained from BeNa Culture Collection (Beijing, China)”. Again, how these strains are related to frozen pork as indicated in the title and presented in the introduction.

The two strains are the main pathogen of pork which the references afforded could be proved this. Because the two strains are the main pathogens of the chilled pork, the inhibition against the two strains by using NE-SAHW could reflect the influence of NE-SAHW on chilled pork preservation.

L107-108: “..and then activated three generations”. What do the authors mean? Please describe adequately every procedure you have employed.

The bacteria were kept in refrigerator and the low temperature makes bacteria dormant or inactive, and it needed to be activated. It is usually necessary for activating three generations that the bacteria would exhibit the best activities. Microbiology experiments always need this step before starting the research. The detail method of activation is using inoculation ring to pick colonies to broth medium and cultivate 24 h, (activate for the first generation), and then pick colonies to broth medium and cultivated 24 h again (activate for the second generation), then replicate for one more time (activate for the third generation).

L111-112: “The resulting pellets were re-suspended in phosphate buffer solution to a final concentration of 10E7, 10E5, 10E2 log CFU/mL”. How this was accomplished?

The resulting pellets were re-suspended in phosphate buffer solution to a final concentration of 107, 105, 102 CFU/mL by plate coating counting method. Using  decimal dilutions and plating on agar surface and incubation, the data was calculated by plate coating counting.

L120: “Reactions were stopped by a terminating agent (0.5% Na2S2O3, 0.85% NaCl)”. Again, where this is coming from?

If there is HOCl and OCl- in solution, Na2S2O3 would react with them first, thus HOCl and OCl- would be consumed up. The reaction between HOCl and bacteria could be stopped. This is the neutralizer against chlorine compound.

L150-151: “OD280 and OD260 value, respectively, with respectively UV spectrophotometer.” This sentence should be deleted.

This sentence has been deleted.

L155: “Determination of bacterial death of two bacteria was assessed using PI-stained flow”. You mean two strains and not just two bacteria. Also, PI stands for propidium iodine. Why don’t you explain that?

This sentence has been corrected: Determination of bacterial death of two bacteria strains was assessed using PI-stained flow. PI stands for propidium iodine. As a fluorescent dye, propidium iodide (PI) can only penetrate through dead/damaged cell and bind with nucleic acid which would exhibit red fluorescence, while it cannot enter inside as the cell membrane remains intact in viable cells which would show no fluorescence (Fröhling et al., 2012). Thus, the amount of PI interaction with nucleic acid signifies cell wall damage or perforation (Rodino et al., 2014).

Fourth are the Results and Discussion (not discussions L 163).

L165-167: “Bactericidal Efficacy of with different level of SAEW and different treatment time on Strains of P. fragi and P. fluorescens were evaluated in this present study respectively and the results were shown in Table 1, 2 and Fig. 2, 3.”

and  L170”.. which could also be verified in Fig. 2, 3..”.

Only numerical figures (data) can support an assumption. Pictures of Petri dishes do not verify anything. The data in Table 1 and 2 are two few to support the findings. These numerical figures also haven’t been compared or statistically verified. How many times did the team repeat the experiments? Where are those data?  Which is the limit of detection for the bacterial enumeration and how did they perform this, with decimal dilutions, plating on agar surface and incubation or the McFarland approach?

Pictures of Petri dishes could not represent any data but representing intuitively the inhibition of different level of NE-SAHW and treatment times against different initial populations of the strains. The data in Table 1 and 2 are the specific data which repeated for three times. The experiment indicated the inhibition of NE-SAHW were related to the level of NE-SAHW, treatment time and initial concentration of the strains. Log reduction was determined with decimal dilutions and plating on agar surface and incubation, the data was calculated by plate coating counting.    

L189 & L494: In figure caption “..Experimental diagram of.. ”. In what way a series of petri dish photos are comprising an experimental diagram? There are proper procedures to analyze photos and extract proper information and data to further explore.

The petri dishes were with different concentration of NE-SHAW against two strains, they exhibited colonies intuitively that reflected the inhibition of NE-SHAW.    

L225. Section 3.3. All of your findings should be supported by numerical data from adequate repetitions of experiments.  Those data should be presented mainly in tables and analyzed statistically to support any differences (or similarities) among the various treatments.

The date presented in figures could be intuitive and intracellular material leakages including protein, nucleic acids and K+ in bacteria could be showed in one figure through double coordinates.

L247. Section 3.4

No data to support any assumption. Just unprocessed FTIR spectra. To the common eye, there are no differences between the control and the treatment with NE-SAHW. How many times have you repeated the analysis?

Actually, there are differences between the control and the treatment, it has been discussing in the manuscript. We have repeated for 3 times.

L 282-296. Section 3.5.

In the text, some percentages are discussed (% of intact cells vs. dead). Where are these data in figures A-D?

99.23% int the text are the blue part (99.81%+0.42%) of figure 8(A), 0.77% in the text are the red part (0.33%+0.44%) of figure 8(A). 42.1% in the text are also the blue part (41.93%+0.17%) figure 8(B), and 57.9% in the text are also the red part (33.61%+24.29%) figure 8(B).